# Essential Fitness Repertoire of *Staphylococcus aureus* during Co-infection with *Acinetobacter baumannii In Vivo*

Gang Li,[a] Wei Shen,[b] Yali Gong,[c] Ming Li,[a] Xiaocai Rao,[a] Qian Liu,[d] Yanlan Yu,[e] Jing Zhou,[a] Keting Zhu,[f] Mengmeng Yuan,[a] Weilong Shang,[a] Yi Yang,[a] Shuguang Lu,[a] Jing Wang,[a] Yan Zhao[a]

aDepartment of Microbiology, College of Basic Medical Sciences, Army Medical University, Chongqing, China

bInstitute for Viral Hepatitis, Department of Infectious Diseases, Key Laboratory of Molecular Biology for Infectious Diseases, Ministry of Education, the Second Affiliated Hospital of Chongqing Medical University, Chongqing, China

cState Key Laboratory of Trauma, Burns and Combined Injury, Chongqing Key Laboratory for Proteomics Disease, Institute of Burn Research, Southwest Hospital, Army Medical University, Chongqing, China

dDepartment of Laboratory Medicine, Ren Ji Hospital, School of Medicine, Shanghai Jiao Tong University, Shanghai, China

eDepartment of Dermatology, Southwest Hospital, Army Medical University, Chongqing, China

fDepartment of Emergency, the Second Affiliated Hospital of Army Medical University, Chongqing, China

Gang Li and Wei Shen contributed equally to this work. Author order was determined both alphabetically and in order of increasing seniority.

**ABSTRACT** *Staphylococcus aureus* represents a major human pathogen that is frequently involved in polymicrobial infections. However, the prevalence and role of co-infectious microbes on the pathogenesis and fitness essentiality of *S. aureus in vivo* remain largely unknown. In this study, we firstly performed a retrospective surveillance of 760 clinical samples and revealed a notable predominance of co-infection with *S. aureus* and *Acinetobacter baumannii*. The high-density *S. aureus* transposon mutant library coupled to transposon insertion sequencing (Tn-Seq) further identified a core set of genes enriched in metabolism of inorganic ions, amino acids, and carbohydrates, which are essential for infection and tissue colonization of *S. aureus* in the murine systemic infection model. Notably, we revealed a differential requirement of fitness factors for *S. aureus* in tissue-specific (liver and kidney) and infection-type-specific manner (mono- and co-infection). Co-infection with *A. baumannii* dramatically altered the fitness requirements of *S. aureus in vivo*; 49% of the mono-infection fitness genes in *S. aureus* strain Newman were converted to non-essential, and the functionality of ATP-binding cassette (ABC) transporters was significantly elicited during co-infection. Furthermore, the number of genes essential during co-infection (503) outnumbers the genes essential during mono-infection (362). In addition, the roles of 3 infection-type-specific genes in *S. aureus* during mono-infection or co-infection with *A. baumannii* were validated with competitive experiments *in vivo*. Our data indicated a high incidence and clinical relevance of *S. aureus* and *A. baumannii* co-infection, and provided novel insights into establishing antimicrobial regimens to control co-infections.

**IMPORTANCE** Polymicrobial infections are widespread in clinical settings, which potentially correlate with increased infection severity and poor clinical outcomes. *Staphylococcus aureus* is a formidable human pathogen that causes a variety of diseases in polymicrobial nature. Co-infection and interaction of *S. aureus* have been described with limited pathogens, mainly including *Pseudomonas aeruginosa*, *Candida albicans*, and influenza A virus. Thus far, the prevalence and role of co-infectious microbes on the pathogenesis and fitness essentiality of *S. aureus in vivo* remain largely unknown. Understanding the polymicrobial composition and interaction, from a community and genome-wide perspective, is thus crucial to shed light on *S. aureus* pathogenesis strategy. Here, our findings demonstrated, for the first time, that a high incidence rate and clinical relevance of co-infection was caused by *S. aureus* and *Acinetobacter baumannii*, illustrating the importance of polymicrobial nature in investigating *S. aureus* pathogenesis. The infection-type-specific genes likely serve as

Address correspondence to Yan Zhao, hnyanyanxp@aliyun.com.

The authors declare no conflict of interest.

potential therapeutic targets to control *S. aureus* infections, either in mono- or co-infection situation, providing novel insights into the development of antimicrobial regimens to control co-infections.

**KEYWORDS** *Staphylococcus aureus*, *Acinetobacter baumannii*, co-infection, Tn-seq, essential fitness

Infections caused by multiple species of microorganisms, also known as polymicrobial infections, are prevalent in the clinical setting and account for approximately 25% of clinical infections (1). Among polymicrobial communities, co-infectious microbes may develop diverse interactions, either mutualistic or competitive, in response to physico-chemical microenvironments and nutrient availability, which ultimately shapes the spatial organization, pathogenic potential, and disease capability of the community (2, 3). For instance, the "food for detoxification" relationship was established for the oral opportunistic pathogen *Aggregatibacter actinomycetemcomitans* and the commensal *Streptococcus gordonii*, where *A. actinomycetemcomitans* spatially colocalizes around, but maintains an optimal distance ($> 4\ \mu$m) from *S. gordonii*, which allows for the metabolic cross-feeding of L-lactate and the simultaneous reduction of peroxide—both of which are produced by *S. gordonii* (2, 4). This mutualistic synergy between the 2 species results in increased bacterial burden and augmented virulence during abscess formation as compared with infection by each single-species alone (2, 4, 5). Similar fine-scale polymicrobial interactions have been depicted in various microbes (6–10), which have not only broadened our understanding of bacterial pathogenesis strategies, but also revealed novel potential interventions contributing to the control and elimination of polymicrobial infections.

*Staphylococcus aureus* is a formidable human pathogen that causes a variety of diseases of polymicrobial nature, such as polymicrobial pneumonia, diabetic foot ulcers, and prosthetic joint infections (11, 12). The co-infectious microbes could serve as specific stress factors, exerting pleiotropic effects on the behavior and fitness of *S. aureus*, which results in altered repertoires related to multispecies competition (13), antibiotic resistance (14, 15), virulence (16, 17), and/or host immune evasion (18). One well-studied model is exemplified by co-infection of *S. aureus* and *Pseudomonas aeruginosa* in the lungs of cystic fibrosis (CF) individuals (19). Adaptation to the CF environment modulates the interaction patterns and elicits either a coexisting or competitive status between *S. aureus* and *P. aeruginosa*. In addition, co-infections and interactions of *S. aureus* with other microbes have also been described, either phenotypically or mechanistically, such as *Candida albicans* (14, 20, 21), influenza A virus (22–25), and even severe acute respiratory syndrome coronavirus 2 (SARS-CoV2) (26–28). Notably, *S. aureus* co-infections have always demonstrated increased infectious severity and poor clinical outcomes. Nonetheless, considering the important clinical relevance and diverse microbes potentially cohabitating with *S. aureus*, the prevalence and effect of co-infection microbes on the pathogenesis and *in vivo* fitness essentiality of *S. aureus* remain poorly understood. Thus, a comprehensive understanding of polymicrobial infections is needed in order to thoroughly investigate *S. aureus* pathogenesis.

In this study, we firstly performed a retrospective surveillance of 760 infection samples recovered from 208 burn patients hospitalized in the intensive care unit, and the infection types as well as microbial compositions were analyzed. Notably, co-infection caused by *S. aureus* and *Acinetobacter baumannii* (a non-fermenting Gram-negative pathogen) predominated in collected samples. Then we utilized transposon insertion mutagenesis coupled with high-throughput sequencing (Tn-seq) to determine the *in vivo* interactions between *S. aureus* and *A. baumannii*. The gene essentiality for *S. aureus* during co-infection with *A. baumannii* was probed at the genome-wide scale using a murine systemic infection model. The results showed that the fitness requirements of *S. aureus* were dramatically altered during co-infection with *A. baumannii*, with 49% of the essential genes needed during mono-infection converted to non-essential during co-infection. Our work illustrates the high incidence rate and clinical relevance of

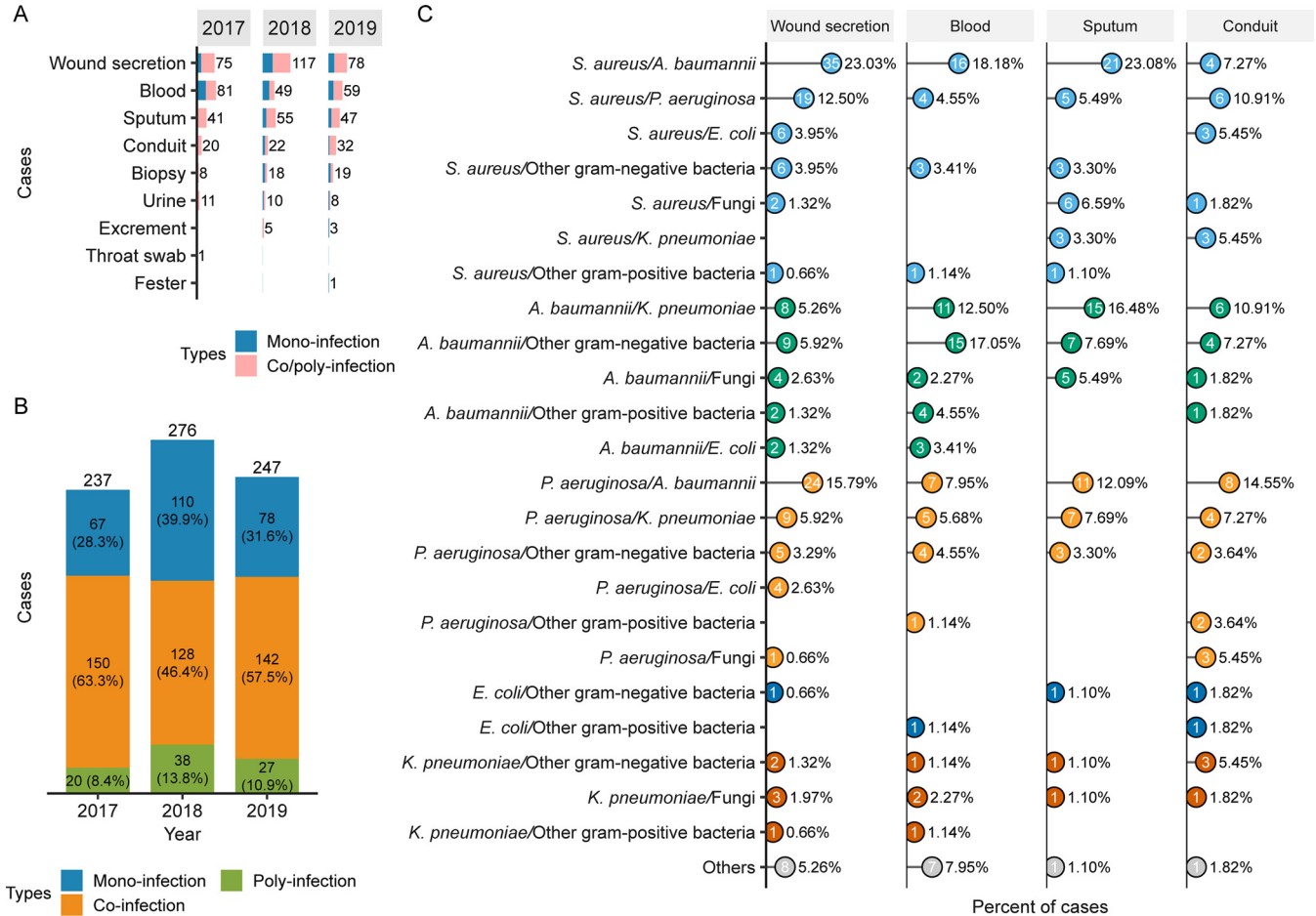

**FIG 1** Statistics of the collected clinical samples and bacterial diversity. (A) A total of 760 clinical samples were recovered from 208 burn patients that were hospitalized in the intensive care unit during 2017 to 2019. The collected samples had diverse origins for both mono-infections and co/poly-infections. (B) The 760 clinical samples were classified into three major types, namely, mono-infection, co-infection, and poly-infection, according to the bacterial species isolated from each sample. Multi-species infections are prevalent in clinical settings, where co-infections dominate the clinical samples. (C) Statistics of the bacterial composition for co-infections according to the sample origins. The co-infecting microbes were assigned to various distinct species, including the antibiotic-resistant ESKAPE pathogens (*S. aureus, P. aeruginosa, A. baumannii, K. pneumoniae*). Co-infection of *S. aureus* and *A. baumannii* showed the highest incidence rates and accounted for 23.03%, 18.18%, and 23.08% of the samples collected from wound secretion, blood, and sputum, respectively.

studying *S. aureus* in polymicrobial infections and provides novel insights into *S. aureus* virulence strategies *in vivo*.

## RESULTS

**Co-infection of *S. aureus* and *A. baumannii* predominated in collected clinical samples.** *S. aureus* represents a common cause of polymicrobial infections (2, 3); however, it is mostly studied in single-species infections. To further survey the infection type and microbial composition of *S. aureus*-related multispecies infections, we first carried out a retrospective study, where 760 clinical samples were collected from various tissues of 208 burn patients who were hospitalized in the intensive care unit during 2017 to 2019. The specimens of various origins, such as wound secretion, blood, sputum, conduit, biopsy, and urine, exhibited preferred multispecies infections versus single-species infections (Fig. 1A). In addition, the samples were further grouped into 3 major categories: mono-infection (caused by single-species), co-infection (caused by 2 different species), and poly-infection (caused by 3 or more different species). The incidence of mono-infection in 2017, 2018, and 2019 was 28.3%, 39.9%, and 31.6%, respectively; the incidence of poly-infection was 8.4%, 13.8%, and 10.9%, respectively (Fig. 1B). In comparison, co-infections predominated in collected samples with an incidence rate of 63.3%, 46.4%, and 57.5% in 2017, 2018, and 2019, respectively

(Fig. 1B). These results indicated that multispecies infections constitute a prevalent situation in clinical settings, which deserves more attention and further investigation.

Out of 760 clinical samples, 420 (55%) were classified into co-infection type. To elucidate the diversity of co-infections, the co-infecting microbial pairs were further analyzed in detail according to the sample origins, including wound secretion, blood, sputum, and conduit (Fig. 1C). Note that the numbers of co-infection samples collected from other origins (biopsy, urine, excrement, throat swab, and fester) were all less than 10, so we did not analyze these samples further. The results showed that the co-infecting microbes were assigned to distinct species, including the antibiotic-resistant *Enterococcus faecium, S. aureus, Klebsiella pneumoniae, A. baumannii, P. aeruginosa, Enterobacter* spp. (ESKAPE) pathogens and/or fungus; whereby the diversified combinations of co-infection were identified (Fig. 1C). Notably, the combination of co-infection with *S. aureus* and *A. baumannii* was mostly recovered and accounted for 23.03%, 18.18%, and 23.08% of the samples collected from wound secretion, blood, and sputum, respectively (Fig. 1C). These data suggested that co-infections predominate in collected samples with potential clinical relevance. Given the highest incidence of co-infection with *S. aureus* and *A. baumannii* identified in this study and the limited understanding of the interaction between the two species, we mainly focused our study on the infections caused by *S. aureus* and *A. baumannii*, while other co-infection combinations also deserve further consideration.

**Construction and characterization of a high-density transposon insertion mutant library in *S. aureus* strain Newman.** Transposon mutagenesis represents a robust tool that enables unbiased genome-wide identification of gene essentiality under specific *in vitro* and/or *in vivo* conditions (29, 30). Here, we firstly constructed a high-density transposon insertion mutant library in *S. aureus* reference strain Newman using the transposon mariner, which enable to insert efficiently into the TA dinucleotides and generates thousands of mutants across a genome (31). Combined with next-generation high-throughput sequencing, the library obtained 171,620 specific insertions that were specifically mapped to the genome of strain Newman, with an average distance of 31-bp between the 2 successive transposon insertions (Fig. 2 and Table S1). *S. aureus* strain Newman has 2,696 putative genes in total, including 2,624 genes coding for proteins and 72 for RNAs. By comparing the transposon insertion frequency across the Newman genome, 174 genes encoding proteins were identified as essential for *S. aureus* survival under *in vitro* conditions based on the absence or underrepresentation of insertions in the transposon library (Table S2). Notably, these genes were functionally enriched for central cellular processes, such as ribosome activity, DNA replication, cell division, and central metabolism (Table S2).

**Analysis of fitness determinants required for *S. aureus* mono-infection.** Infectious studies were conducted to screen for fitness determinants required for *S. aureus* mono-infection using a murine model. BALB/c mice were inoculated intravenously via the tail vein with the Newman transposon library ($1 \times 10^7$ CFU per mouse). The livers and kidneys of the infected mice at 5 days post-injection were recovered, homogenized, and prepared for Tn-seq. K-means clustering of the principal-component analysis (PCA) highlighted that the Tn-seq data generated from input and mono-infection clustered independently from each other, both in the tissue colonization of livers and kidneys (Fig. 3A and B). To specifically focus on the *in vivo* essential factors, the *in vitro* essential genes (174) were removed from the transposon input pool, and excluded from all subsequent analyses.

Among the 2,522 genes that were non-essential for *S. aureus* viability *in vitro*, a total of 362 genes were identified essential for *S. aureus* mono-infection (Fig. 3C and Table S2). Specifically, 26 (7%) genes encode fitness factors required for the colonization of both the liver and kidney, representing a core set of genes essential for *S. aureus in vivo* mono-infection, 265 (73%) genes encoded liver-specific fitness factors, and 71 (20%) were kidney-specific (Fig. 3C and Table S2). Furthermore, 326 mono-infection essential genes could be annotated according to the classification of the gene category within the Clusters of Orthologous Groups (COG) database. In addition to the COG category of function unknown, these genes were mostly enriched in the COG categories for amino acid, inorganic ion, and carbohydrate metabolism, as well as cell envelope biogenesis (Fig. 3D).

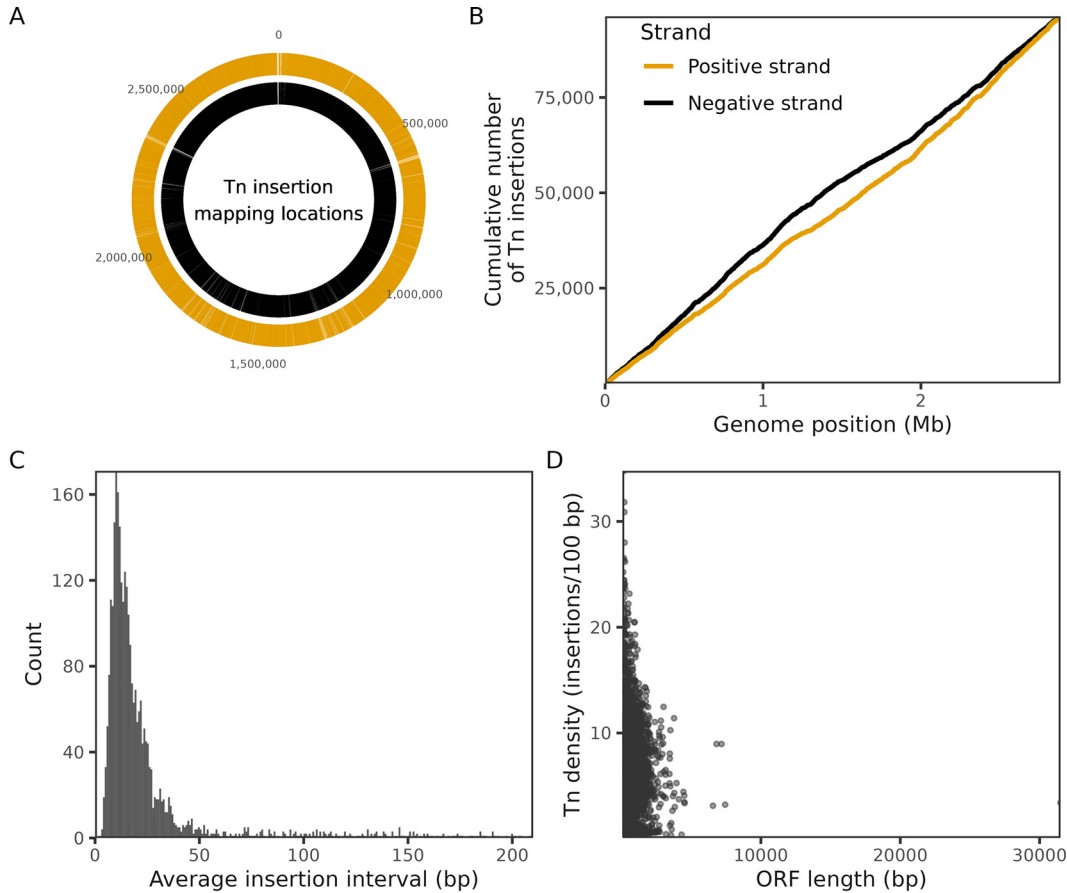

**FIG 2** Quality assessment of the constructed Tn insertion mutant library. (A) Distribution of the genome-wide Tn insertion locations mapped to the genome of *S. aureus* strain Newman. The outer ring (red) represents the positive strand, and the inner ring (blue) represents the negative strand. (B) Cumulative number of Tn insertions in positive strand (red) and negative strand (blue). (C) Statistics of the Tn insertion number per interval (bp). The peak indicates an average distance of 31-bp between the two successive transposon insertions. (D) Statistics of the Tn density (number of insertions per 100-bp).

**Analysis of fitness determinants required for *S. aureus* co-infection with *A. baumannii*.** Given that *S. aureus* and *A. baumannii* were the most commonly co-isolated microbes in various infection samples in this study, we next assessed the fitness factors crucial for *S. aureus* during the co-infection with *A. baumannii*. A mixture of Newman transposon insertion library and 10-fold less *A. baumannii*, which attempts to avoid the unexpected effect caused by increased inoculum, was inoculated into mice. Liver and kidney samples were collected and subjected to Tn-seq analysis. The data of co-infection exhibited distinct PCA clustering from that of both the input and mono-infection (Fig. 3A and B), indicating that the presence of *A. baumannii* alters the gene essentiality for *S. aureus*.

Tn-seq identified 503 genes in total that were essential for *S. aureus* co-infection with *A. baumannii in vivo* (Fig. 3C and Table S2), which outnumber the genes essential during mono-infection (362). Further examination revealed that 421 (84%) genes were essential for *S. aureus* colonization of the liver, 53 (10%) genes specific for kidney, and only 29 (6%) genes were required for colonization of both tissues (Fig. 3C and Table S2), indicating that distinct subsets of *S. aureus* genes were required for different tissue colonization. Similarly, the genes essential for co-infection were mostly enriched in the COG functional categories of metabolism for amino acid, inorganic ion, carbohydrate, as well as the biological process for transcription and cell envelope biogenesis (Fig. 3D).

**Presence of *A. baumannii* dramatically alters the fitness requirements for *S. aureus in vivo*.** Given the limited information regarding the *in vivo* interplay between *S. aureus* and *A. baumannii*, we further assessed how the presence of *A. baumannii*

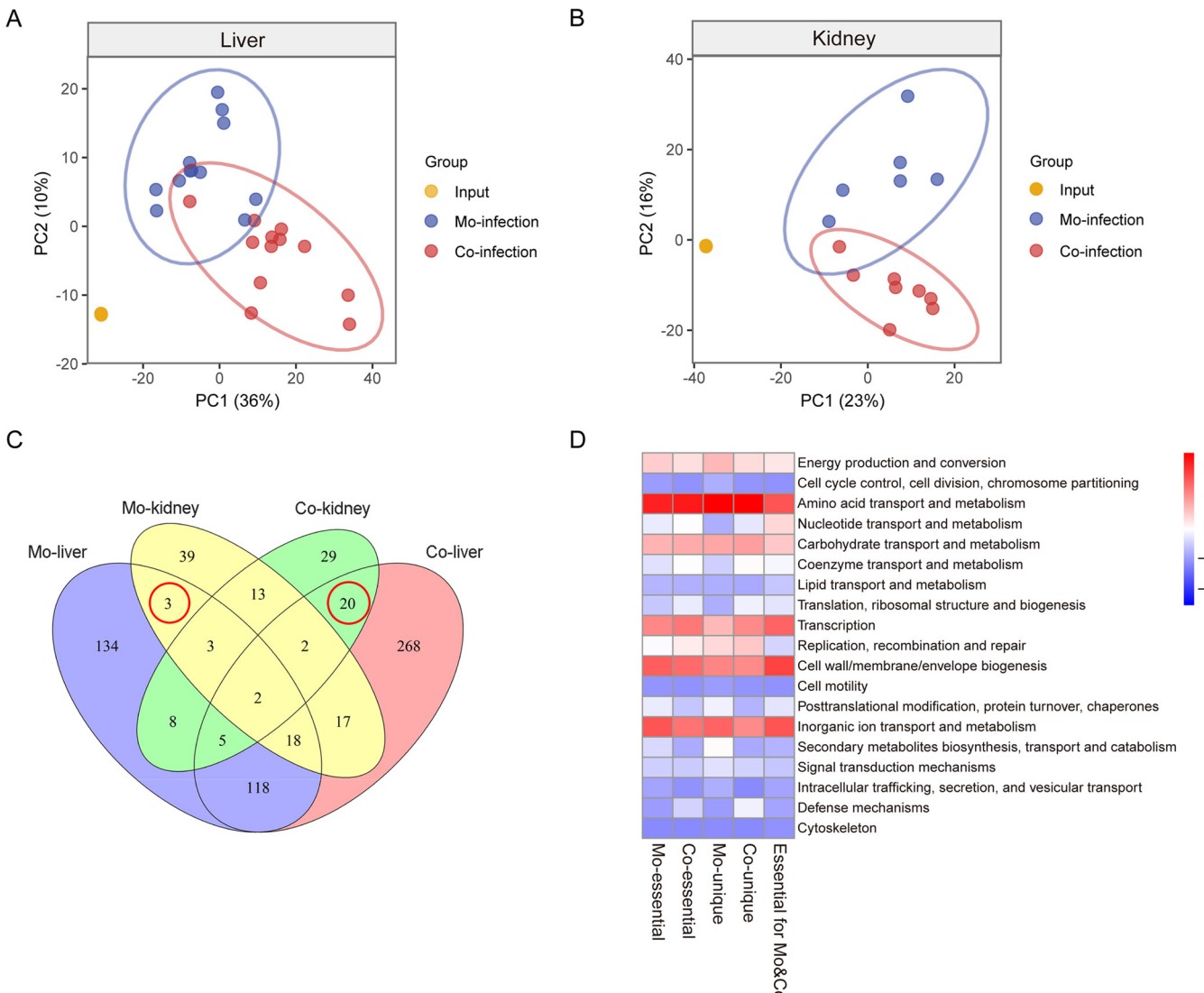

**FIG 3** Clustering and functional annotation of Tn-seq data. (A) Principal-component analysis of the normalized Tn-seq counts recovered from the liver in three conditions: input (orange, *n* = 4, the points are overlapped), mono-infection (blue, *n* = 12), and co-infection (red, *n* = 12). (B) Principal-component analysis of the normalized Tn-seq counts recovered from the kidney in three conditions: input (orange, *n* = 4, the points are overlapped), mono-infection (blue, *n* = 6), and co-infection (red, *n* = 8). (C) Venn diagram of the essential genes of *S. aureus* required for *in vivo* colonization of the liver and kidney both in mono-infection and co-infection with *A. baumannii*. The red circle depicts the genes essential for colonization of both the liver and kidney while also being divergently required for mono-infection or co-infection. (D) Heatmap of COG functional categories of the essential *in vivo* genes of *S. aureus*. Conserved COG functionality was assigned to both infection conditions, while differentiated requirements were observed for genes unique to mono-infection and co-infection conditions.

impacts the fitness requirements of *S. aureus* in a murine systemic infection model. Notably, integrated analysis of the data recovered from both mono-infection and co-infection conditions revealed that 176 (26%) genes were required specifically for *S. aureus* mono-infection (designated hereinafter as genes unique to mono-infection), 317 (47%) genes were required exclusively for co-infection condition (designated hereinafter as genes unique to co-infection), and 186 (27%) genes were essential for both infection types (Fig. 3C and Table S2). Three genes unique to mono-infection were required for colonization of both the liver and kidney (Fig. 3C), including the gene encoding ornithine cyclodeaminase SbnB (NWMN_0061), competence protein ComGB (NWMN_1447), and hypothetical protein (NWMN_0560). Twenty genes unique to co-infection, such as phosphotransferase system protein TreP (NWMN_0438), oxidoreductase (NWMN_2478), deoxynucleoside kinase (NWMN_0518), Na$^+$/H$^+$ antiporter MnhG (NWMN_0599), and oligopeptide transport permease (NWMN_0856), were required for the colonization of both tissues (Fig. 3C).

Interestingly, genes encoding known virulence-associated factors, including exotoxins, cofactors and enzymes, adhesins, biofilm, and global regulators, exhibited distinct essentiality during mono-infection and co-infection conditions. Several factors were required for both infection types, such as the staphylococcal accessory gene regulator AgrC, fibrinogen-binding protein ClfA, biofilm-associated N-glucosaminyltransferase IcaA, sortase SrtA, and iron-regulated heme-iron binding protein IsdA (Table S2). In contrast, co-infection with *A. baumannii* elicited additional requirement of several virulence factors, particularly for the two-component system members (SaeR, ArlR, HssS, and KdpE), cysteine protease SspB, and hyaluronate lyase HysA (Table S2). These results indicated that the essential fitness factors of *S. aureus* were dramatically changed during co-infection with *A. baumannii*, and about 49% of the essential genes (176) for *S. aureus* mono-infection were converted to non-essential in the presence of *A. baumannii*. Meanwhile, a total of 317 non-essential genes during mono-infection were additionally required during co-infection. The transition of essential genes between mono-infection and co-infection conditions depicts an interesting reciprocal relationship.

Furthermore, 186 genes essential for *S. aureus* across both infection types represented a core set of genes and were mostly enriched in the COG functional categories of transport and metabolism of inorganic ions, amino acids, carbohydrates, and nucleotides (Fig. 3C and D), elucidating a critical role of metabolism for *S. aureus* during *in vivo* infection. In addition to shared metabolism activities, the COG annotation of infection-type-specific fitness genes resulted in divergent functional enrichments for co-infection compared to mono-infection conditions, mainly exhibiting the overrepresentation of the COG category of nucleotide and coenzyme metabolism, translation, and defense mechanisms, together with the alleviation of secondary metabolites catabolism (Fig. 3D).

Kyoto Encyclopedia of Genes and Genomes (KEGG) pathways analysis revealed further insights into the distinct functions of *in vivo* essential genes either unique to mono-infection or to co-infection. Notably, KEGG pathways of lysine degradation and base excision repair predominated genes specific for mono-infection (Fig. 4), suggesting these functions might be compensated by *A. baumannii* during co-infection. Of note, 5 genes involved in lysine degradation were identified, which include genes encoding aldehyde dehydrogenase AldA (NWMN_0113), dihydrolipoamide succinyltransferase SucB (NWMN_1325), 2-oxoglutarate dehydrogenase SucA (NWMN_1326), d-alanine aminotransferase (NWMN_1643), and acetyl-CoA C-acetyltransferase VraB (NWMN_0539) (Fig. 4). Four genes were identified to be related to base excision repair, including endonuclease III (NWMN_1363), DNA-3-methyladenine glycosidase (NWMN_1559), formamidopyrimidine-DNA glycosylase (NWMN_1582), and DNA ligase (NWMN_1842) (Fig. 4). Additionally, fitness factors unique to co-infection were significantly enriched in the KEGG pathway of ATP-binding cassette (ABC) transporters ($P = 0.02$) (Fig. 4). Consistently, 27 genes encoding ABC transporters, which potentially participate in the transportation of diverse substrates, such as peptides, phosphate, biotin, siderophore, heme, and bacitracin, were screened out of the 317 genes unique to co-infection (Fig. 4). The overrepresentation of ABC transporters for genes specific to co-infection suggested that the presence of *A. baumannii* elicited additional functions, i.e., nutrients competition and/or resilience toward harmful antibacterial agents, required for *S. aureus* to successfully establish *in vivo* infection and tissue colonization.

**Validation of candidate fitness factors required in type-specific infection.** To verify the accuracy of the genome-scale Tn-seq analyses, 3 individual genes, including the mono-infection-unique essential gene *sbnB* (NWMN_0061) encoding ornithine cyclodeaminase, the co-infection-unique essential gene *treP* (NWMN_0438) encoding trehalose-specific transporter, and the *in vivo* non-essential gene *sasF* (NWMN_2545) encoding cell-wall-anchored protein SasF, were selected based on the fold change of Tn-seq reads of differentially screened genes. Note that gene *sbnB* and *treP* was required for distinct infection types but essential for colonization of both the liver and kidney, allowing to assess in detail the gene essentiality in specific infection-type manner. Then, the corresponding mutants (Δ*sbnB*, Δ*treP*, and Δ*sasF*) were generated in *S. aureus* Newman background. Bacterial growth curve

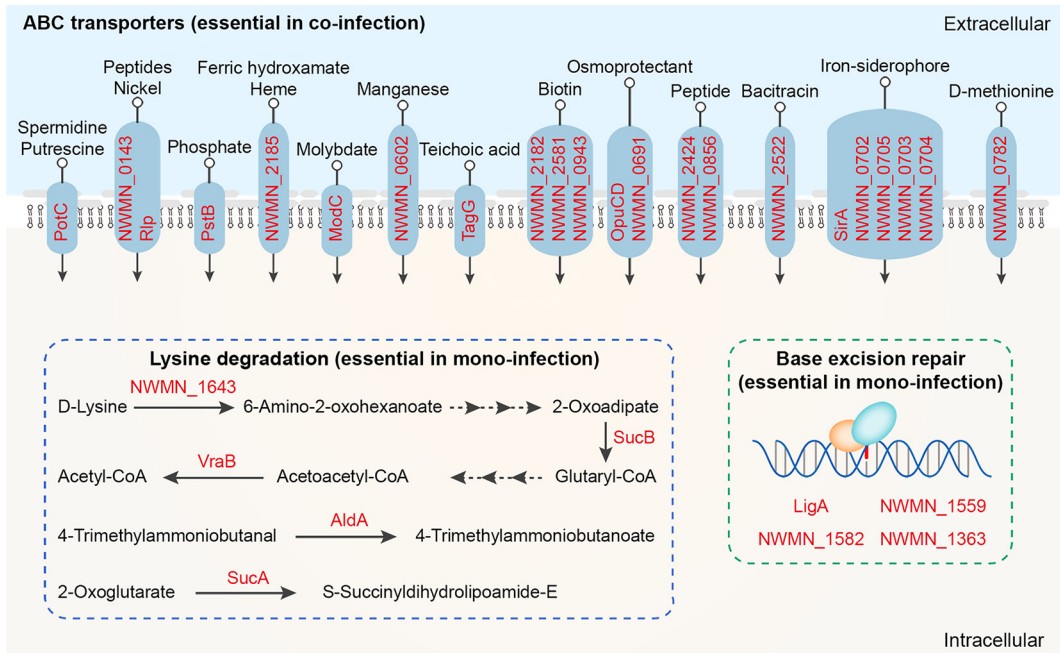

**FIG 4** Functionality analysis of essential *S. aureus in vivo* genes unique to mono-infection and co-infection. Schematic diagram of the functionality of enriched genes (labeled as red color). The genes required exclusively for mono-infection were mostly enriched for KEGG pathways of lysine degradation (*P*-adj value of 0.06) and base excision repair (*P*-adj value of 0.20). Co-infection-unique genes were significantly enriched in KEGG pathway of ABC transporters (*P*-adj value of 0.02). The genes encoding ABC transporters participate in the transportation of diverse substrates. Note that 5 genes with unknown substrate specificity, including NWMN_1203, NWMN_1758, NWMN_0251, NWMN_0654, NWMN_2340, were not drawn in the diagram.

determination revealed that all 3 mutants exhibited similar growth patterns compared to the wild-type (WT) strain Newman (Fig. 5A). We then co-infected each of the mutants against strain Newman with or without *A. baumannii* in the murine systemic model. The livers and kidneys were collected 5 days postinfection and processed for viable CFU counts of *S. aureus*. As expected, mutation of gene *sasF* resulted in no significant difference in the CFU recovered from both tissues between Δ*sasF* and WT, regardless of the presence of *A. baumannii* (Fig. 5B). Decreased colonization in both the liver and kidney was observed when mutant Δ*sbnB* competed against WT, exclusively in mono-infection but not in co-infection with *A. baumannii* (Fig. 5B). By contrast, gene *treP* was crucial for efficient colonization of *S. aureus* in the liver and kidney during co-infection with *A. baumannii*, but not required for mono-infection condition (Fig. 5B).

To gain further insights into the differential requirement of the gene *treP* for *S. aureus* infection *in vivo*, mice were inoculated with WT or mutant Δ*treP*, either as a single-species injection or co-injected with *A. baumannii*, and the survival rates of mice were monitored. No obvious difference regarding the survival of the mice was found between the WT and Δ*treP* (Fig. 5C). In comparison, the presence of *A. baumannii* augmented the virulence of the WT strain, while it significantly alleviated the virulence of the mutant Δ*treP*, which led to increased and prolonged mouse survival under tested conditions (Fig. 5C). Together, these results showing direct *in vivo* co-infection with the fitness gene mutant and WT strain Newman confirmed the infection-type-specific phenotypes, which were consistent with the primary Tn-seq results in this study.

## DISCUSSION

Polymicrobial infections have been historically recognized (2, 9, 32); however, the detailed interactions and pathogenesis strategies between different species remain largely unknown. *S. aureus* is a major human pathogen that is frequently involved in polymicrobial infections, and the presence of co-infectious microbes complicates the *in vivo* behaviors of

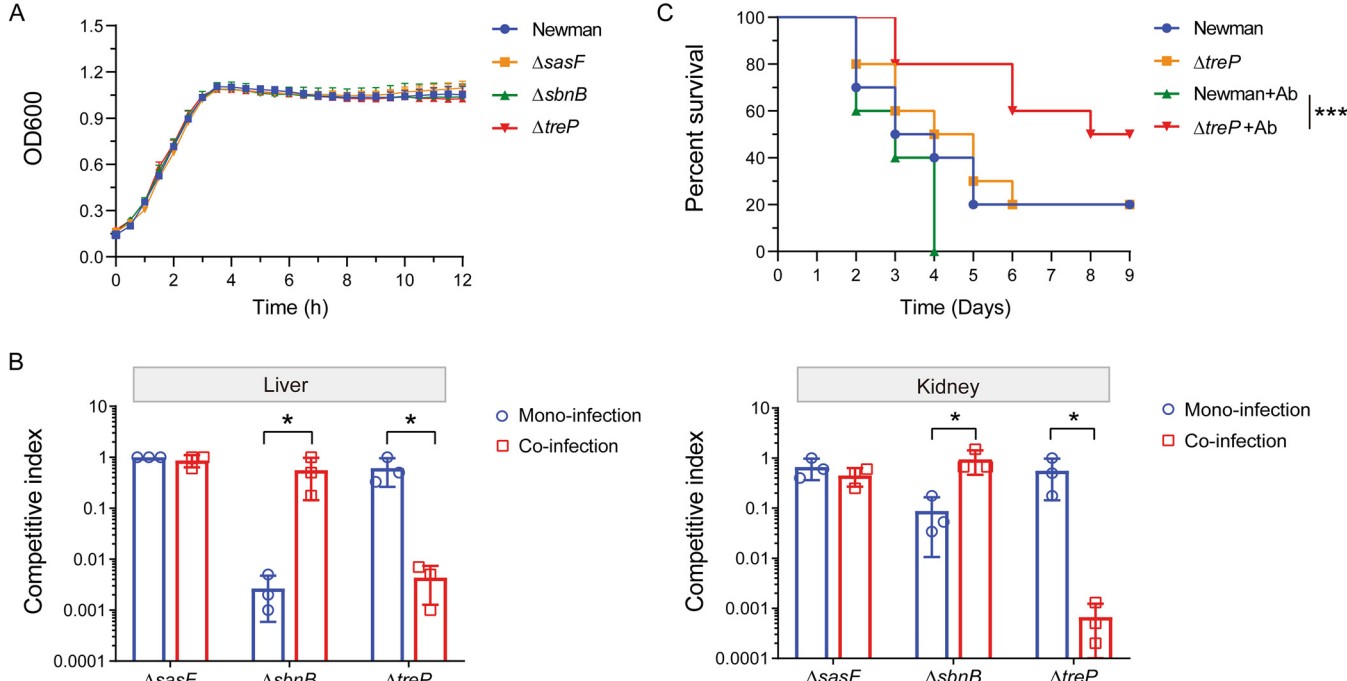

**FIG 5** Validation of selected fitness factors for *S. aureus in vivo* pathogenesis. (A) *In vitro* growth monitoring of *S. aureus* strain Newman and three gene mutants. Note that gene *sasF* is non-essential for *S. aureus in vivo* infection, gene *sbnB* is required for mono-infection, and *treP* is required for co-infection. No obvious difference was observed in the growth of the mutants compared to the WT strain Newman. (B) *In vivo* competition assay. The 3 infection-specific gene mutants were used in competition assays with the WT strain Newman in mono- and co-infection with *A. baumannii* in a murine systemic infection model. Bacterial CFU were recovered from both the livers and kidneys 5 days postinfection, and the competitive index for each gene mutant was calculated using WT as the control. The results are presented as the mean ± SD (standard deviation) from 3 biological replicates. Statistical analysis was performed using a Student's t test, *, $P < 0.05$. (C) Mice survival curves. Mice ($n = 10$) were inoculated via tail vein injection with a bacterial suspension at $2 \times 10^7$ CFU of either WT or mutant, or $2 \times 10^7$ *S. aureus* and $2 \times 10^6$ *A. baumannii* for co-infection. Survival of the mice was monitored for 9 days post-injection. Ab, *A. baumannii*. Statistical analysis was performed by log-rank (Mantel-Cox) test, ***, $P < 0.001$.

*S. aureus* compared to mono-infection conditions. To date, the majority of studies regarding co-infections and interactions between *S. aureus* and other microbial community members have focused on *S. aureus* and *P. aeruginosa* (19, 33), as well as the pathogenic fungus *C. albicans* (14, 20), highlighting the importance of polymicrobial infections in investigating *S. aureus* pathogenesis. In this study, we discovered a widespread incidence of co-infections caused by *S. aureus* and *A. baumannii* through a retrospective surveillance of clinical samples recovered from burn patients. The results of this study broaden our knowledge of the diverse species that cause co-infections with *S. aureus*.

Despite the high prevalence of both *S. aureus* and *A. baumannii* in the same infection niches, the interplay between the two species has been poorly explored. A previous study cocultured clinical strains of *A. baumannii* and *S. aureus* that were recovered from the same soft tissue of a diabetic patient and found that these strains exhibit a state of commensalism *in vitro*, without effects on each other either beneficially or detrimentally (34). Another study demonstrated that the presence of *A. baumannii* alters the pharmacodynamics and modulates the killing of *S. aureus* by meropenem *in vitro*, pointing out that dose-optimized beta-lactams represent a therapeutic option for control of co-infections involving *S. aureus* and *A. baumannii* (35). Here, we firstly investigated the fitness requirement for *S. aureus* to establish infection and colonize tissue *in vivo*. Co-infection with *A. baumannii* dramatically altered the fitness essentiality of *S. aureus* by increasing the co-infection-unique genes (317) and alleviation of the mono-infection-unique genes (176). Specifically, approximately 49% of the mono-infection essential genes converted to non-essential during co-infection. Similarly, a previous study showed via a murine model of chronic surgical wound infection that co-infection with *P. aeruginosa* results in conversion of ~25% of the monoculture essential genes in *S. aureus* to non-essential (36), revealing the complicated interactions between *S.*

*aureus* and *P. aeruginosa in vivo*. In addition, while the transition of essential genes between mono-infection and co-infection was observed in the presence of *A. baumannii*, the cues and underlying mechanism eliciting this transition remain to be fully explored. We speculate that the differential requirement of essential genes might derive from the direct interaction between the two species, but also raise the possibility that host factors function in this transition, such as limited nutrients might indirectly shape the interaction. Further integrated analysis of the altered essential genes *in vivo* and the direct interaction of *S. aureus* and *A. baumannii in vitro* may contribute to the comprehensive understanding of this crucial process.

In this study, 186 genes were identified as *in vivo* fitness determinants for *S. aureus* during both single-species infection and co-infection with *A. baumannii*, suggesting that these genes likely encode functions crucial for *S. aureus* pathogenesis and tissue colonization. Among them, the core set of fitness factors was mostly assigned to the COG functionality of metabolism of inorganic ions, amino acids, carbohydrates, and nucleotides, coinciding with findings of other studies where central metabolism plays a critical role for *S. aureus* infection *in vivo* (36, 37). In addition, 176 genes identified as unique to *S. aureus* mono-infection were converted to non-essential during co-infection with *A. baumannii*, and these genes were mostly enriched in KEGG pathways of lysine degradation and base excision repair. Notably, metabolism of lysine has been shown to have diverse effects on *S. aureus* physiology, resistance, and pathogenesis. For example, modification of membrane lipids with L-lysine confers *S. aureus* with resistance to human defensins and evasion of neutrophil killing (38), and increased lysine amounts have been shown to benefit adaptation of *S. aureus* after internalization by human lung epithelial cells (S9 and A549) and human embryonic kidney cells (HEK293) (39). Furthermore, a widespread bacterial lysine degradation pathway that converts glutarate to succinate provides an important link in central carbon and energy metabolism (40). Thus, an interesting question is how the presence of *A. baumannii* contributes to the lysine degradation of *S. aureus* during co-infection. The specific impact on *S. aureus* virulence deserves further investigation.

The ability of *S. aureus* to withstand damage caused by the host immune defense is crucial for the successful establishment of an infection, in which DNA is a common target of host-producing antimicrobials. A previous study demonstrated that neutrophils cause DNA double-strand breaks in the genome of *S. aureus* through reactive oxygen species produced by the oxidative burst, and such DNA damage can be repaired by an AddAB-family of helicase/nuclease complexes (RexAB) (41). The repair of DNA damage enables the survival of *S. aureus* in host tissues during infection, indicating that DNA damage repair represents an important mechanism by which *S. aureus* withstands host defense (41). In this study, 4 genes involved in the functionality of base excision repair were identified unique to *S. aureus* mono-infection, but were not required for co-infection with *A. baumannii*, suggesting that *A. baumannii* may provide functional compensation for *S. aureus* to avoid and/or cope with the DNA damage encountered within the same infection microenvironment. However, the mechanisms underlying this relationship are unclear and need further investigation.

Another interesting finding is that among the 317 genes essential exclusively during co-infection, a total of 27 genes encoding transporters were significantly enriched in the KEGG pathway of ABC transporters ($P = 0.02$). Moreover, these ABC transporters potentially participate in the transportation of a broad spectrum of substrates, including peptides, phosphate, biotin, siderophore, heme, and bacitracin. Similar results were obtained in a previous study showing that co-infection with *S. gordonii* imposes unique metabolic stresses on *A. actinomycetemcomitans* as multiple nutrient transporters were solely required during co-infection (36). Membrane transport systems are abundant membrane proteins in *S. aureus*, with niche- and environment-specific roles in sustaining cell integrity and metabolic homeostasis, contributing to bacterial growth, antibiotic resistance, and virulence (42). The additional requirement of ABC transporters for *S. aureus* during co-infection with *A. baumannii* may be explained by 2 aspects, namely,

competition for limited nutrients in a specific niche and/or efflux of harmful factors generated by *A. baumannii*.

Furthermore, this study revealed differential requirements of fitness factors for *S. aureus* colonization in the liver and kidney. Among the 679 genes essential for *S. aureus* infection *in vivo*, 520 (77%) genes were unique to colonization of the liver, 81 (12%) were unique to kidneys, and only 2 genes (NWMN_0574 and NWMN_0639) were required for colonization in both tissues and mono/co-infection types. A Tn-seq screen using a murine model of catheter-associated urinary tract infection revealed distinct fitness requirements for the opportunistic pathogen *Providencia stuartii* colonization in either the bladder or kidneys during single-species infection or co-infection with a common co-colonizer, *Proteus mirabilis* (43). A previous study identified a LysR-type transcriptional regulator (LTTR) required for efficient colonization of *S. aureus* in the kidney, but not in the liver, in a murine model of metastatic bloodstream infection (44). The requirement of different subsets of genes as depicted in this study and previous reports highlights the remarkable plasticity of *S. aureus* to coordinate its expression of fitness factors, which in turn contribute to adaptation to the niche-specific microenvironment and nutrient availability during infection.

In addition, since the first report of Tn mutagenesis in *S. aureus* using the *Enterococcus faecalis* transposon Tn917 in 1997 (45), Tn-seq has emerged as a powerful high-throughput technique that is widely applied to identify infection-specific fitness determinants of *S. aureus* (29). Given the nature of the transposons mostly used, several common limitations remain to be addressed, including inability to accurately assess the fitness contribution of secreted factors, mainly due to cross-complementation, which demonstrated, in our study and others, that genes encoding exoproducts were poorly identified in Tn-seq screens (29, 36). In general, with the introduction of improved Tn methodologies, including a bacteriophage-based transposition system (46) and CRISPR-Cas12k-based, RNA-guided mutagenesis (47), Tn-seq may contribute to the future understanding of the physiology, adaptation, and evolution of *S. aureus*, such as the key factors responsible for *S. aureus* virulence in dynamic environments (e.g., as a time series, or with specific niche relevance), transmission from livestock to humans, and genetic mutations or phenotypic variations critical for vancomycin-intermediate resistance.

Overall, this study revealed a high incidence rate and important clinical relevance of co-infections caused by *S. aureus* and *A. baumannii*. The presence of *A. baumannii* dramatically alters the fitness determinants essential for *S. aureus* infection *in vivo*, particularly the functional compensation of lysine degradation and DNA repair, as well as the additional requirement of factors encoding ABC transporters. The exact mechanism by which the presence of co-infecting *A. baumannii* leads to altered fitness requirements will be an interesting subject of future studies. Moreover, the infection-type-specific genes contributing to *S. aureus in vivo* pathogenesis likely represent promising targets for the development of potential therapeutic interventions aimed at controlling both single-species and polymicrobial infections.

## MATERIALS AND METHODS

**Bacterial strains and culture conditions.** Bacterial strains and plasmids used in this study are listed in Table S3. Clinical samples were procured with approval from the Ethics Committee of the First Affiliated Hospital of Army Medical University, PLA, under the ethics statement number KY201991. Clinical samples were collected from various tissues of 208 burn patients hospitalized in the intensive care unit of the Institute of Burn Research of Southwest Hospital of Army Medical University from years 2017 to 2019. The recovered bacteria were further identified using matrix-assisted laser desorption/ionization time-of-flight mass spectrometry (MALDI-TOF MS). Repetitive samples with the same microbes from the same tissue of the same patient were excluded, and samples with the same microbes from different sites of the same patient were included. Unless otherwise specified, *S. aureus* strain Newman and *A. baumannii* strain ATCC19606 used in this study was cultured in brain heart infusion broth (BHI) and LB broth at 37°C with shaking at 200 rpm, respectively.

**Construction of the transposon insertion library in *S. aureus* strain Newman.** The plasmid MA15 possesses a temperature-sensitive replicon and a mariner transposon, with a kanamycin resistance gene and an MmeI restriction site within each inverted repeat, and was used to construct the transposon insertion mutant library in the *S. aureus* strain Newman as described previously (48). Approximately 60,000 transposon insertion colonies were collected and stored in 20% glycerol at −80°C. Total DNA

was extracted from the library freezer stock using the TIANamp Bacteria DNA Kit (Tiangen) according to the manufacturer's recommendations, and subjected to Tn-Seq libraries preparation as the input.

**Murine systemic infection.** Murine systemic infections were performed with 6–8 week old female BALB/c mice via tail vein injection. The mice were purchased from HUNAN SJA laboratory animal Co., LTD (Hunan, China) and housed in two-way housing on a 12-h light/dark cycle. All animal experiments were performed in strict accordance with protocols approved by the Laboratory Animal Welfare and Ethics Committee of the Third Military Medical University and complied with Institutional Animal Welfare and Ethical guidelines (AMUWEC2019421).

The preparation of the mutant library for injection was performed as previously described (49). Briefly, a 100 $\mu$L aliquot of the mutant library freezer stock was inoculated into 100 mL of BHI without antibiotic selection and cultured overnight at 37°C. To isolate cells in exponential growth phase, the overnight culture was inoculated 1:1000 into fresh BHI without antibiotics, and bacterial growth was monitored until an optical density of 600 nm (OD600) of 0.8 was achieved. For Tn-seq experiments, $1 \times 10^7$ CFU of the *S. aureus* Newman transposon mutant library was used for mono-infection ($n = 12$ mice), and a mixture of $5 \times 10^6$ CFU of the *S. aureus* library and $5 \times 10^5$ CFU of the *A. baumannii* strain ATCC19606 were used for co-infection ($n = 12$ mice). Here, 10-fold less *A. baumannii* than *S. aureus* was used in the inoculum to ensure that the effects observed were not due to increased bacterial inoculum as previously described (36).

Five days postinfection, the livers and kidneys were collected from the infected mice and snap-frozen in liquid nitrogen. The liver and kidney tissues were then ground carefully into a powder in liquid nitrogen, resuspended in TE buffer in Sarstedt 2 mL microtubes prefilled with 1 g of 0.1 mm zirconia beads (BioSpec), and homogenized with a Mini-Beadbeater (Biospec). The tissue lysates were then subjected to genomic DNA extraction using the TIANamp Bacteria DNA Kit (Tiangen).

**Preparation of Tn-seq libraries for sequencing.** Genomic DNA extracted from the livers and kidneys of mice subjected to the mono- or co-infection murine systemic infection models were prepared for Tn-seq analysis as described previously (30). Briefly, genomic DNA was digested with MmeI (Thermo Scientific), treated with calf intestinal alkaline phosphatase (NEB), and purified by the Wizard SV Gel and PCR Clean-Up System (Promega). Then, a double-stranded adapter, generated by annealing the primers of 5'-phosphorylated oligonucleotide A and oligonucleotide B (Table S4), was ligated to the purified DNA fragments. The DNA fragments were then subsequently amplified using the primer pair P1-transposon-MmeI and P2-tn-seq-PCR (Table S4), in which the barcodes for multiplexing were added simultaneously, and a 103-bp sequence with 20-bp of bacterial-derived DNA was obtained. The PCR products were separated on a 2% agarose gel, extracted with the gel purification column (Promega), and then processed via high-throughput sequencing using an Illumina Hi-Seq platform (Novogene). After sequencing, different samples were distinguished based on barcode sequence, and the 20-bp bacterial-specific sequences were mapped to the genome of strain Newman. The distribution of Tn insertions across the entire genome was calculated.

**Tn-seq bioinformatics analysis.** Raw Tn-seq reads were cleaned with fastp v0.20.1 (50) and demultiplexed with SeqKit v0.13.0 (51). The 20-bp genomic sequences were extracted using SeqKit and mapped to the genome of *S. aureus* strain Newman using SeqKit. In-house scripts were used to filter unique mapped sequence with at most one mismatch. Gene mutations were summarized with Bedtools v2.29.2 (52). Essential genes were determined with DESeq2 v1.30.1 (53) with an adjusted $P$ value $< 0.05$ and $\log_2$ (fold change) $< -1$. KEGG pathway enrichment was analyzed by clusterProfiler v3.14.3 (54) with an adjusted $P$ value $< 0.05$.

**Construction of *S. aureus* gene mutants.** For mutation of genes *sasF*, *treP*, and *sbnB* in the strain Newman, approximately 1000-bp regions upstream and downstream of the specific genes were amplified from Newman genomic DNA. The KanR cassette-encoding gene was amplified using the plasmid MA15 as the template (primer sequence in Table S4). The 3 DNA fragments in the upstream-KanR cassette-downstream orientation were then cloned into the temperature-sensitive plasmid pBT2 via Gibson assembly (NEB). Knock-in mutations of the target genes replaced by the KanR cassette were generated via homologous recombination as previously described (55), which were then ultimately verified by sequencing.

**Validation of mutant fitness using the *in vivo* systemic infection model.** Log-phase cultures of *S. aureus* strain Newman or the KanR cassette knock-in mutants were washed, normalized to an equal OD600 reading in PBS, and mixed at a ratio of 1:1 of mutant to WT. Mice ($n = 5$ per group) were inoculated via tail vein injection with a mixture of $1 \times 10^7$ CFU for mono-infection or $5 \times 10^6$ CFU *S. aureus* and $5 \times 10^5$ CFU *A. baumannii* for co-infections. Both the livers and kidneys were recovered 5 days postinfection and homogenized with a tissue grinder (SCIENTZ-48L, SCIENTZ). The homogenate was 10-fold serially diluted in PBS and plated on BHI agar (to quantify all *S. aureus*) and BHI agar supplemented with 100 $\mu$g/mL kanamycin (to quantify *S. aureus* mutants). For co-infection conditions, both *S. aureus* and *A. baumannii* formed colonies on BHI agar but could be easily differentiated by colony color due to the yellow pigment staphyloxanthin secreted by *S. aureus*. Competitive indices for each mutant in each infection type were calculated as ([CFU of mutant *S. aureus*/tissue]/[CFU of WT *S. aureus*/tissue]).

For mouse survival monitoring, log-phase cultures of *S. aureus* strain Newman and the individual fitness gene mutant were washed and normalized to an equal OD600 reading in PBS, respectively. Mice ($n = 10$ per group) were inoculated via tail vein injection with $2 \times 10^7$ CFU of either WT or mutant or $2 \times 10^7$ CFU *S. aureus* and $2 \times 10^6$ CFU *A. baumannii* for co-infection. Growth and survival of mice were monitored for 9 days post-injection.

**Statistics.** Unless noted otherwise, data were analyzed by GraphPad Prism v8.0 (GraphPad Software Inc.). Significant differences were determined by a Student's *t* test for comparison of two independent

data sets or with one-way analysis of variance (ANOVA) followed by Tukey's multiple comparison test for multiple comparisons. A *P* value of less than 0.05 was considered significant.

**Data availability.** The raw Tn-seq data have been deposited into the NCBI Sequence Read Archive under accession number PRJNA787005. The genome sequence and annotation file of *S. aureus* strain Newman is available under GenBank accession number AP009351.

## SUPPLEMENTAL MATERIAL

Supplemental material is available online only.

**TABLE S1**, XLSX file, 0.01 MB.
**TABLE S2**, XLS file, 0.2 MB.
**TABLE S3**, DOCX file, 0.02 MB.
**TABLE S4**, DOCX file, 0.02 MB.

## ACKNOWLEDGMENTS

This work was funded by the National Natural Science Foundation of China (grant 31900145 to G.L.), the Chongqing Municipal Natural Science Foundation of China (grant cstc2019jcyj-msxmX0142 to Y.Z.), and the Science Foundation of the Army Medical University (grant 2019JCLC02 to M.L.).

We thank LetPub (www.letpub.com) for linguistic assistance and pre-submission expert review. No potential conflict of interest was reported by the authors.

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
