## [Reviewer comments · mSystems]

Essential fitness repertoire of *Staphylococcus aureus* during co-infection with *Acinetobacter baumannii* *in vivo*

Gang Li, Wei Shen, Yali Gong, Ming Li, Xiancai Rao, Qian Liu, Yanlan Yu, Jing Zhou, Keting Zhu, Mengmeng Yuan, Weilong Shang, Yi Yang, Shuguang Lu, Jing Wang, and Yan Zhao

Corresponding Author(s): Yan Zhao, Third Military Medical University

Review Timeline:

Submission Date:	April 7, 2022
Editorial Decision:	May 24, 2022
Revision Received:	June 8, 2022
Editorial Decision:	July 24, 2022
Revision Received:	August 8, 2022
Accepted:	August 13, 2022

Editor: Gilles van Wezel

Reviewer(s): The reviewers have opted to remain anonymous.

Transaction Report:

DOI: <https://doi.org/10.1128/msystems.00338-22>

May 24, 2022

Dr. Yan Zhao
Third Military Medical University
Department of Microbiology
gaotanyan street #30
chongqing 400038
China

Re: mSystems00338-22 (Essential fitness repertoire of *Staphylococcus aureus* during co-infection with *Acinetobacter baumannii* *in vivo*)

Dear Dr. Yan Zhao:

Thank you for submitting your manuscript to mSystems. We have completed our review and find the paper of interest for our journal. Acceptance will depend on whether you can adequately address the reviewer comments. The paper has been seen by two expert reviewers, the comments of whom you will find below. While both have indicated "minor revisions", in their comments (and direct communication to the editor) they also indicate that conclusive explanations for treatment differences are missing, leaving a significant part to hypothesis. Please address this as well as possible in the new draft.

Preparing Revision Guidelines

Sincerely,

Gilles van Wezel

Editor, mSystems

Journals Department
Reviewer comments:

Reviewer #1 (Comments for the Author):

In the paper titled: Essential fitness repertoire of *Staphylococcus aureus* during co-infection with *Acinetobacter baumannii* in vivo` Gang Li et al describe changes in the role of certain genes of *S. aureus* during mono-, and co infection with *A. baumannii*.

The paper addresses the important clinical aspect of co-infection, a matter currently underexposed. With a large set of data the authors approach the subject, yet several unfortunate choices were made. In general the paper is well written though at several places not crystal clear.

In the abstract was mentioned: `Co-infection with *A. baumannii* dramatically altered the fitness requirements of *S. aureus* in vivo; 49% of the mono-infection fitness genes in *S. aureus* strain Newman were converted to non-essential`. This reviewer thinks that despite this observation, the finding that the number of genes essential during coinfection ($n=503$) still outnumber the genes essential during mono infections ($n=362$) should be mentioned in the abstract as well.

In this study 1,058 clinical samples were included obtained from burn patients who were hospitalized in the intensive care unit of the Institute of Burn Research of Southwest Hospital of Army Medical University during 2017 to 2019. Unfortunately it was not mentioned from how many patients the samples originated and in how many of them the different combinations of mono-, coinfections etc. were observed.

The authors use an *S. aureus* murine systemic infection model for their experiments which is a pity as a burn wound infection model would have been more appropriate.

Unfortunately it is not clear what is meant by: differentially screened, (see following passage) and what is depicted in figure 4. `Interestingly, genes encoding known virulence-associated factors, including exotoxins, cofactors and enzymes, adhesins, biofilm, and global regulators, were differentially screened in mono- and co-infection conditions. Several factors were conserved for both two infection types, such as the staphylococcal accessory gene regulator AgrC, fibrinogen-binding protein ClfA, biofilm-associated N-glucosaminyltransferase IcaA, sortase SrtA, and iron-regulated heme-iron binding protein IsdA (Fig. 4).`

At several places like for instance in the following sentence, it is not really clear what is meant with: genes unique to co-infection. `Another interesting finding is that among the genes unique to co-infection, a total of 27 genes encoding transporters were (sentence 366).`

The authors mention that per infection 12 mice were used, but it is not clear how much variation was found between the different mice within one group.

The use of strain Newman is for obvious reasons of accessibility a good choice but to extrapolate data found with this lab strain to *S. aureus* in general is questionable. Confirmation experiments in recent clinical isolates would have been more appropriate.

Reviewer #2 (Comments for the Author):

The aim of this paper is to use TnSeq to identify essential genes in *Staphylococcus aureus* during mouse infection when co-inoculated with or without *Acinetobacter baumannii*. The study first uses a retrospective analysis of clinical samples from burn patients to determine the fraction of infections that contained either 1, 2 or more species (mono, co-, or polyinfections). Infections across three years were dominated by co- or poly- infections, including 14% of co-infections with *S. aureus* and *A. baumannii*, a pairing that the authors (somewhat arbitrarily) used as their focus. Next, they use a saturated Tn library with TnSeq to identify essential genes during liver and kidney (co)/infection. As expected, some genes are essential and these differ depending on the presence of *A. baumannii*. Results for the fitness effects of 2 arbitrary genes are confirmed using strains with targeted knock-outs. The paper is clear and well-written. However, there is scant analysis of the results that would provide any clear indication of what the essential genes might be doing in the context of either mono- or co-infection. The observations/speculations may lead to testable hypotheses, but these aren't discussed in much detail.

Some specific questions and comments are given below.

- 1) The retrospective analysis overstates the high frequency of S.a. and A.b. co-occurrence (lines 144-45). Several pairs occur essentially as often. And it may be, based on their rates of mono-infection, that their co-occurrence is what would be expected from chance. This should be easily estimated, at least for the most common species.
 - a. In Figure 1, and lines 130-135, I'd recommend distinguishing mono from co+poly-infections. The key is that Sa is frequently found with other species. It is completely fine, and undoubtedly important, to focus to focus on Sa + Ab, but it should be more clearly stated that this is 1 of many options that can be considered with this Tn library.
 - b. In 1C, please indicate the number of cases and not just the frequencies. Also, the colors are not specified; the colors can probably be removed, because they create confusion with the same colors already used in 1A and 1B.
- 2) The results indicate that there are more essential genes during co-infection (503) than during mono-infection (362). The authors emphasize in several locations (e.g. abstract and lines 231-234) that 49% of the genes that are essential during mono-infection were converted to non-essential during co-infection. Two things:
 - a. There is little further discussion of why this transition occurs. Why do these essential genes become non-essential?
 - b. As interesting as this may be, I think the reciprocal is even more interesting, that non-essential genes during mono-infection become essential during co-infection. As above, which genes and are there ideas for their importance?
 - c. In both cases above, there is the implication that the "cause" is the direct interaction between the two species. It is equally possible that the effects are mediated entirely via the host. It would have been very informative control to have included in vitro experiments with Sa and Ab to complement the in vivo experiments. This limitation should be noted and discussed.
- 3) Survival data in Fig 6c are based on a very small sample size per mutant (5 mice/strain). This is too few to allow any meaningful conclusions, however suggestive. These conclusions should be toned down considerably given the limitations of the experiments.
 - a. For the same experiments, please justify why these particular mutants were chosen.
- 4) Functional interpretations are generally lacking in the paper, even for the few groups that are given special attention (ABC transporters, DNA excision repair, lysine degradation). Why these pathways? For the transporters, is the hypothesis that co-infection requires more import or export?
- 5) Fig 4 can be removed. It isn't discussed in the text.
- 6) Fig 5: I'd suggest adding a legend to know which pathways are essential in which conditions (mono or co-infection).

In the paper titled: Essential fitness repertoire of *Staphylococcus aureus* during co-infection with *Acinetobacter baumannii* in vivo` Gang Li et al describe changes in the role of certain genes of *S. aureus* during mono-, and co infection with *A. baumannii*.

The paper addresses the important clinical aspect of co-infection, a matter currently underexposed. With a large set of data the authors approach the subject, yet several unfortunate choices were made. In general the paper is well written though at several places not crystal clear.

In the abstract was mentioned: `Co-infection with *A. baumannii* dramatically altered the fitness requirements of *S. aureus* in vivo; 49% of the mono-infection fitness genes in *S. aureus* strain Newman were converted to non-essential`. This reviewer thinks that despite this observation, the finding that the number of genes essential during coinfection (n=503) still outnumber the genes essential during mono infections (n=362) should be mentioned in the abstract as well.

In this study 1,058 clinical samples were included obtained from burn patients who were hospitalized in the intensive care unit of the Institute of Burn Research of Southwest Hospital of Army Medical University during 2017 to 2019. Unfortunately it was not mentioned from how many patients the samples originated and in how many of them the different combinations of mono-, coinfections etc. were observed.

The authors use an *S. aureus* murine systemic infection model for their experiments which is a pity as a burn wound infection model would have been more appropriate.

Unfortunately it is not clear what is meant by: differentially screened, (see following passage) and what is depicted in figure 4. `Interestingly, genes encoding known virulence-associated factors, including exotoxins, cofactors and enzymes, adhesins, biofilm, and global regulators, were differentially screened in mono- and co-infection conditions. Several factors were conserved for both two infection types, such as the staphylococcal accessory gene regulator *AgrC*, fibrinogen-binding protein *ClfA*, biofilm-associated N-glucosaminyltransferase *IcaA*, sortase *SrtA*, and iron-regulated heme-iron binding protein *IsdA* (Fig. 4).`

At several places like for instance in the following sentence, it is not really clear what is meant with: genes unique to co-infection. `Another interesting finding is that among the genes unique to co-infection, a total of 27 genes encoding transporters were (sentence 366).`

The authors mention that per infection 12 mice were used, but it is not clear how much variation was found between the different mice within one group.

The use of strain Newman is for obvious reasons of accessibility a good choice but to extrapolate data found with this lab strain to *S. aureus* in general is questionable. Confirmation experiments in recent clinical isolates would have been more appropriate.

Yan Zhao, PhD

Associate professor of Department of Microbiology, Army Medical University, Chongqing, 400038, China. Tel.: 86-23-68752243 E-mail: hnyanyanxp@aliyun.com

Reviewer

mSystems

Ref.: Research article (mSystems00338-22)

Dear reviewer:

On behalf of my co-authors, we thank you very much for giving us an opportunity to revise our manuscript entitled “**Essential fitness repertoire of *Staphylococcus aureus* during co-infection with *Acinetobacter baumannii* in vivo**” (mSystems00338-22). We appreciate you very much for your positive and professional comments and suggestions on our manuscript. We have studied your comments carefully and have made revision which **marked in tracked changes** in the revised manuscript. Here are point-by-point answers to the concerns.

Reviewer #1 comments

1. In the abstract was mentioned: “Co-infection with *A. baumannii* dramatically altered the fitness requirements of *S. aureus* in vivo; 49% of the mono-infection fitness genes in *S. aureus* strain Newman were converted to non-essential”. This reviewer thinks that despite this observation, the finding that the number of genes essential during coinfection (n=503) still outnumber the genes essential during mono infections (n=362) should be mentioned in the abstract as well.

Response: Thanks for your precious suggestions. The result of the number of genes essential during co-infection outnumber that of during mono-infection has been included in the abstract in the revised manuscript (please see line 38-39).

2. In this study 1,058 clinical samples were included obtained from burn patients who

were hospitalized in the intensive care unit of the Institute of Burn Research of Southwest Hospital of Army Medical University during 2017 to 2019. Unfortunately it was not mentioned from how many patients the samples originated and in how many of them the different combinations of mono-, coinfections etc. were observed.

Response: Thanks for your comments. In fact, 1,058 clinical samples from 208 burn patients were collected during 2017 to 2019. Among them, 298 cases originated from 85 patients were collected in 2017, 419 cases from 53 patients were collected in 2018, 341 cases from 70 patients were taken in 2019. Due to that samples would be collected from the same patient varied in origins including wound secretion, blood, sputum, conduit, biopsy, and urine, the different infection types were analyzed based on the 1,058 cases other than 208 patients. We have added the number of patients in the revised manuscript (please see line 106).

3. The authors use an *S aureus* murine systemic infection model for their experiments which is a pity as a burn wound infection model would have been more appropriate.

Response: Thanks for your professional comments. In this study, the clinical samples were collected from burn patients, but varied in the origins including wound secretion, blood, sputum, conduit, biopsy, and urine. Despite of the local infections, majority of the burn patients exhibited systemic infection and/or deep tissue infection. Thus, we applied the murine systemic infection model in this study to assess the gene essentiality for *S. aureus* mono-infection and co-infection with one of the mostly recovered opportunistic bacterium *A. baumannii*, in which the potential fitness factors crucial for tissue (liver and kidney) colonization could be identified simultaneously. Nonetheless, the burn wound infection also represents an ideal model to profile the genes essential for local burn wound infection, and it will be of major consideration to our future studies.

4. Unfortunately it is not clear what is meant by: differentially screened, (see following passage) and what is depicted in figure 4. “Interestingly, genes encoding known virulence-associated factors, including exotoxins, cofactors and enzymes,

adhesins, biofilm, and global regulators, were differentially screened in mono- and co-infection conditions. Several factors were conserved for both two infection types, such as the staphylococcal accessory gene regulator AgrC, fibrinogen-binding protein ClfA, biofilm-associated N-glucosaminyltransferase IcaA, sortase SrtA, and iron-regulated heme-iron binding protein IsdA (Fig. 4).”

Response: Sorry for the imprecise description. We have rephrased this sentence in the revised manuscript as the following: genes encoding known virulence-associated factors exhibited distinct essentiality during mono-infection and co-infection conditions. Several factors were required for both two infection types (please see line 230-233). In addition, Figure 4 has been removed from the revised manuscript to avoid the possible misunderstanding.

5. At several places like for instance in the following sentence, it is not really clear what is meant with: genes unique to co-infection. “Another interesting finding is that among the genes unique to co-infection, a total of 27 genes encoding transporters we (sentence 366).”

Response: Sorry for the misleading description. In our previous draft, the genes required specifically for *S. aureus* mono-infection were recognized as the genes unique to mono-infection; genes required exclusively for *S. aureus* co-infection with *A. baumannii* were recognized as the genes unique to co-infection. In the revised manuscript, we have included and specified the terms of genes unique to mono-infection and genes unique to co-infection (please see line 217-220). Besides, the relevant sentences has been rephrased in the revised manuscript to make it clear.

6. The authors mention that per infection 12 mice were used, but it is not clear how much variation was found between the different mice within one group.

Response: As a result of multiple factors, including infection micro-environment, DNA extraction, library preparation, and sequencing, variations within one group did exist. In the principal component analysis of the normalized Tn-seq counts, the sample variances of PC1 and PC2 in each group were different. For the liver, the

values were 68.2 and 33.3 in the Mo-infection group and 129.0 and 31.0 in the Co-infection group; for the kidney, the values were 91.6 and 85.8 in the Mo-infection group and 61.9 and 29.5 in the Co-infection group. The variances of PC1 and PC2 in the Input groups of liver and kidney were all smaller than 1.0.

7. The use of strain Newman is for obvious reasons of accessibility a good choice but to extrapolate data found with this lab strain to *S. aureus* in general is questionable. Conformation experiments in recent clinical isolates would have been more appropriate.

Response: Thanks for your constructive suggestion. *S. aureus* strain Newman is a laboratory reference strain that has been widely used for studies of *S. aureus* biology and pathogenesis, as in the case of virulence genes previously identified by Tn-seq technique in strain Newman (Bae, T. *et al.* Proc. Natl. Acad. Sci. U. S. A. 2004). Given the updated knowledge in the genome-wide scale and the high efficiency of genetic manipulation of strain Newman, we firstly constructed a high-density transposon insertion mutant library in this strain using the transposon mariner, which was thus further used in the following experiments. Nonetheless, the use of *S. aureus* clinical isolates to profile the gene essentiality provides a distinct perspective and alternative. The integrated analysis and comparison of the results derived from lab strains and clinical isolates might contribute to the advanced knowledge of *S. aureus* pathogenesis with specific clinical relevance, and this will be of our future subject. We have included relevant discussion in the revised manuscript (please see line 434-435). Thanks for your understanding.

Reviewer #2 comments

1. The retrospective analysis overstates the high frequency of S.a. and A.b. co-occurrence (lines 144-45). Several pairs occur essentially as often. And it may be, based on their rates of mono-infection, that their co-occurrence is what would be expected from chance. This should be easily estimated, at least for the most common species.

a. In Figure 1, and lines 130-135, I'd recommend distinguishing mono from co+poly-infections. The key is that Sa is frequently found with other species. It is completely fine, and undoubtedly important, to focus to focus on Sa + Ab, but it should be more clearly stated that this is 1 of many options that can be considered with this Tn library.

b. In 1C, please indicate the number of cases and not just the frequencies. Also, the colors are not specified; the colors can probably be removed, because they create confusion with the same colors already used in 1A and 1B.

Response: Thanks for your professional suggestions. Multispecies infections predominate in various clinical settings. To obtain a comprehensive understanding of this scene, we collected a total of 1,058 clinical samples and analyzed the infectious causes. As indicated, co-infections were widely distributed in our analysis, which exhibited diversified combinations, including co-infections caused by *S. aureus* and *P. aeruginosa*, by *S. aureus* and *E. coli*, by *A. baumannii* and *K. pneumoniae*, by *P. aeruginosa* and *A. baumannii*, by *P. aeruginosa* and *K. pneumoniae*, etc. Notably, co-infections with *S. aureus* and *A. baumannii* were mostly recovered from our analysis. Given the expected clinical relevance and the limited knowledge of interactions between *S. aureus* and *A. baumannii*, we thus focus on the co-infections of the two species in this study. Nonetheless, other co-infection combinations also could be analyzed using the Tn-seq technique, in the case of the well-studied interaction between *S. aureus* and *P. aeruginosa*, and the poor understanding of interaction between *A. baumannii* and *K. pneumoniae*. In the revised manuscript, we have included the description that other co-infection combinations also deserve further consideration (please see line 147-151).

In Figure 1A, to assess the origin of specimens, the clinical samples were distinguished mono-infections from co/poly infections. In Figure 1B, to obtain a detailed classification of the infection types, we thus categorized the samples into three types (mono-infection, co-infection, and poly-infection). In the revised manuscript, we have changed graph with different colors (Figure 1A and 1B), and rephrased the relevant sentences depicting the infection types and incidence (please

see line 131-134).

For Figure 1C, the number of cases has been added as suggested. In addition, we have changed this graph to a distinct type (lollipop chart) with colors different from Figure 1A and 1B, making it more clear. Many thanks.

2. The results indicate that there are more essential genes during co-infection (503) than during mono-infection (362). The authors emphasize in several locations (e.g. abstract and lines 231-234) that 49% of the genes that are essential during mono-infection were converted to non-essential during coinfection. Two things:

a. There is little further discussion of why this transition occurs. Why do these essential genes become non-essential?

b. As interesting as this may be, I think the reciprocal is even more interesting, that non-essential genes during mono-infection become essential during coinfection. As above, which genes and are there ideas for their importance?

c. In both cases above, there is the implication that the "cause" is the direct interaction between the two species. It is equally possible that the effects are mediated entirely via the host. It would have been very informative control to have included *in vitro* experiments with Sa and Ab to complement the *in vivo* experiments. This limitation should be noted and discussed.

Response: Thanks for your professional comments. Co-infection with *A. baumannii* dramatically altered the fitness essentiality of *S. aureus in vivo* by increasing the co-infection-unique genes (317) and alleviation of the mono-infection-unique genes (176). In detail, approximately 49% of the mono-infection essential genes converted to non-essential during co-infection, while the number of genes essential during co-infection (503) outnumbers the genes essential during mono-infection (362). To unveil the potential functionality crucial for this transition, we firstly annotated the identified essential genes with COG database, and revealed divergent functional COG enrichments. Compared to genes essential during mono-infection, the essential genes during co-infection were mainly enriched in the COG categories of nucleotide and coenzyme metabolism, translation, and defense mechanisms, while secondary

metabolites catabolism was overrepresented for mono-infection essential genes. KEGG pathway analysis further revealed possible explanation for this transition, whereby pathways of lysine degradation and base excision repair dominated genes specific for mono-infection, and ABC transporters were significantly enriched in genes essential for co-infection. Actually, several candidate genes were recently tested for this transition at either protein regulatory level or at the small non-coding RNA regulatory level, while the studies remain uncompleted. In the revised manuscript, we have included relevant description and discussion of the reciprocal relationship (please see line 243-246).

Originally, our study aimed to profile the genes essential solely during infection condition, which requires the specific infection site microenvironments. We thus removed the genes essential during *in vitro* growth conditions before subsequent analysis, and compared the essential genes identified during mono-infection and co-infection *in vivo*. The potential factors contributing to this transition thus mainly derive from the direct interaction between the two species. However, we are still unable to exclude the possibility that host factors function in this transition, such as limited nutrients might indirectly shape the interaction. Direct interaction of *S. aureus* and *A. baumannii in vitro* provides an alternative that may contribute to the comprehensive understanding of this crucial process. This limitation has been noted and discussed in the revised manuscript as suggested (please see line 347-355).

3. Survival data in Fig 6c are based on a very small sample size per mutant (5 mice/strain). This is too few to allow any meaningful conclusions, however suggestive. These conclusions should be toned down considerably given the limitations of the experiments.

a. For the same experiments, please justify why these particular mutants were chosen.

Response: The animal experiments were conducted in three independent replicates with five mice per group, and showed consistent results. During mono-infection, the WT and mutant $\Delta treP$ led to similar mice survival. In comparison, co-infection with *A. baumannii* augmented the virulence of the WT, but alleviated the virulence of the

mutant $\Delta treP$, resulting in increased and prolonged mouse survival. In Figure 6C, the data were expressed from one representative of the three independent replicates, and the results are reproducible. Nonetheless, we have toned down the conclusions as suggested in the revised manuscript (please see line 307-310).

For validation of candidate fitness factors, mutants were selected and constructed mainly based on the fold change of Tn-seq reads. Genes that have been reported to participating in *S. aureus in vivo* pathogenesis were excluded. In addition, Genes that are required for distinct infection types but essential for colonization of both the liver and kidney were considered, allowing to assess in detail the gene essentiality in specific infection-type manner. In the revised manuscript, the consideration for mutant chosen has been supplemented (please see line 285-289). Many thanks.

4. Functional interpretations are generally lacking in the paper, even for the few groups that are given special attention (ABC transporters, DNA excision repair, lysine degradation). Why these pathways? For the transporters, is the hypothesis that co-infection requires more import or export?

Response: Sorry for our poor command of the functional interpretations. Using Tn-seq screen, we identified numerous infection-specific genes that might be important for *S. aureus in vivo* pathogenesis; however, the underlying mechanism crucial for this process remains complex. To obtain a general understanding, we performed both COG annotation and KEGG pathway enrichment analysis, revealing possible explanations for the essential genes transition. Notably, the KEGG pathways of lysine degradation and base excision repair predominated genes specific for mono-infection, and ABC transporters were significantly enriched for co-infection. Thus, we mainly focused on these pathways in this study. In the revised manuscript, we have included the functional interpretations as possible (please see line 276-279, 394-396), thanks for your understanding.

For genes solely required during co-infection, 27 genes encoding ABC transporters were screened. This finding is interesting but sophisticated, given the broad spectrum of substrates that could be transported by these transporters, including

peptides, phosphate, biotin, siderophore, heme, and bacitracin. We speculate that the additional requirement of ABC transporters during co-infection with *A. baumannii* might benefit *S. aureus* for limited nutrients competition, and/or increase *S. aureus* resilience towards harmful antibacterial agents produced by *A. baumannii*. These interesting scenarios are under consideration in our recent works. We also have added relevant description in the revised manuscript (please see line 275-279). Many thanks.

5. Fig 4 can be removed. It isn't discussed in the text.

Response: This figure has been deleted in the revised version as suggested.

6. Fig 5: I'd suggest adding a legend to know which pathways are essential in which conditions (mono or co-infection).

Response: Thanks for your precious suggestion. This concern has been addressed in the updated figure in the revised manuscript.

We have tried our best to improve the manuscript and hope we have addressed all the questions raised. We appreciate your earnest work, and hope that the corrections will meet with approval.

Once again, thank you very much for your professional comments and suggestions.

Looking forward to hearing from you.

Best regards.

Yours sincerely,

Yan Zhao, PhD

Department of Microbiology,

Army Medical University,

30# Gaotanyan St., Shapingba District,

Chongqing 400038, China

Tel: 86-23-68752243

E-mail: hnyanyanxp@aliyun.com

July 24, 2022

Dr. Yan Zhao
Third Military Medical University
Department of Microbiology
gaotanyan street #30
chongqing 400038
China

Re: mSystems00338-22R1 (Essential fitness repertoire of *Staphylococcus aureus* during co-infection with *Acinetobacter baumannii* in vivo)

Dear Dr. Yan Zhao:

Thank you for submitting your manuscript to mSystems. We have completed our review of your manuscript. The work is of interest for mSystems and many of the queries were addressed appropriately. While Referee #1 recommended acceptance, Referee #2 noted a major issue that I agree requires attention. The data in Fig 1 report >1000 values, but these are from only 208 patients. Frequencies report overall values, but these are non-independent because multiple samples are taken from the same patients across different sites. This must be addressed in the paper. In particular, it should be made clear how much variation there is between the on average 5 samples per patient (which are highly likely representing the same strain) and the strains between patients. Additional statistics should be performed to address this issue.

Please realise that while I have chosen minor revisions versus reject and allow resubmission, I deem properly addressing this important issue as essential, and we usually do not allow a third revision.

Below you will find instructions from the mSystems editorial office and comments generated during the review.

Preparing Revision Guidelines

Sincerely,

Gilles van Wezel

Editor, mSystems

Journals Department
Reviewer comments:

Reviewer #2 (Comments for the Author):

The authors have generally done a nice job responding to most of my concerns. However, two issues remain that need attention:

- 1) The retrospective analysis of co-infection is from >1000 samples, but these come from only 208 patients. Samples from the same patients are obviously non-independent, even if they are from different sites. This problem potentially affects every value in Figure 1 and essentially invalidates these analyses. Are patients equally represented by multiple samples? Are the same sites sampled more than once through time? Are values correlated across the same patient through time? If this is to be salvaged, I'd suggest limiting the data to only wound sites. It will then still be important to deal with correlated values from the same patients/sites across years (i.e you're inflating the overall frequency of Sa/Ab co-infections if these are present in the same patients through time). I'm sorry I didn't notice this in the original manuscript.
- 2) Although the response indicates that the infections studies in Fig 5C (formerly 6C) are based on 3 independent replicates of 5 mice per treatment, this isn't clear from the text. In fact, the text and legend both repeatedly note that 5 mice were used. Are the plots based on 5 mice or 15 mice? Is there variance across the replicates? If these haven't been analyzed, then why have 3 independent replicates? This needs to be clarified. If the values are indeed only based on 5 mice/treatment, my original concern remains that these are too few to be conclusive.

Yan Zhao, PhD

Associate professor of Department of Microbiology, Army Medical University, Chongqing, 400038, China. Tel.: 86-23-68752243 E-mail: hnyanyanxp@aliyun.com

Reviewer

mSystems

Ref.: Research article (mSystems00338-22)

Dear reviewer:

On behalf of my co-authors, we thank you very much for giving us an opportunity to revise our manuscript entitled “**Essential fitness repertoire of *Staphylococcus aureus* during co-infection with *Acinetobacter baumannii* in vivo**” (mSystems00338-22). We appreciate you very much for your professional comments and suggestions on our manuscript. We have studied your comments carefully and have made revisions which **marked in tracked changes** in the revised manuscript. Here are point-by-point answers to the concerns.

Reviewer #2 comments

1. The authors have generally done a nice job responding to most of my concerns. However, two issues remain that need attention:

1) The retrospective analysis of co-infection is from >1000 samples, but these come from only 208 patients. Samples from the same patients are obviously non-independent, even if they are from different sites. This problem potentially affects every value in Figure 1 and essentially invalidates these analyses. Are patients equally represented by multiple samples? Are the same sites sampled more than once through time? Are values correlated across the same patient through time? If this is to be salvaged, I'd suggest limiting the data to only wound sites. It will then still be important to deal with correlated values from the same patients/sites across years (i.e you're inflating the overall frequency of Sa/Ab co-infections if these are present in the

same patients through time). I'm sorry I didn't notice this in the original manuscript.

Response: Thanks for your professional comments and suggestions. Due to the complexity and prolonged hospital stay for patients that hospitalized in the intensive care unit, samples are usually collected from various tissues and different time points in clinical settings, in which repetitive sampling might exist in practice (Gong YL, *et al. Front Cell Infect Microbiol.* 2021, 11: 681731; Polemis M, *et al. Euro Surveill.* 2020, 25(34): 1900516). To resolve this concern, we re-analyzed the 1,058 samples in detail and excluded the repetitive samples, in which samples with the same microbes from the same tissue of the same patient were excluded, while samples with the same microbes from different sites of the same patient were included. In addition, samples from the same site but differed in infecting microbes were not excluded, which might indicate secondary infection during hospitalization, as in the case of wound infection. After cleaning, a total of 760 samples were ultimately included for further analysis. In the revised manuscript, we have renewed the sample numbers as well as infection incidence rates (please see line 27, 105, 125-135, etc), and provided a revised version for Fig. 1A and Fig. 1B.

Considering that samples from different sites of the same patient might confuse the subsequent statistics of co-infection incidence rates overall. We also re-classified the co-infecting microbes based on sample origins according to the reviewer's suggestions. Note that the numbers of co-infection samples collected from biopsy, urine, excrement, throat swab, and fester are all less than 10, we thus did not analyze these samples further. In detail, the co-infecting microbial pairs were analyzed based on origins, including wound secretion, blood, sputum, and conduit. The results showed that co-infection with *S. aureus* and *A. baumannii* exhibited the highest incidence rates and accounted for 23.03%, 18.18%, and 23.08% of the samples collected from wound secretion, blood, and sputum, respectively. For conduit samples, co-infection with *S. aureus* and *A. baumannii* ranked the fourth that represents a leading cause. In the revised manuscript, we have rephrased the relevant descriptions (please see line 140-144, 149-151, 783-786), and renewed the Fig. 1C.

Overall, the conclusion that co-infection with *S. aureus* and *A. baumannii*

predominated in collected co-infecting samples is convincing. Many thanks.

2. Although the response indicates that the infections studies in Fig 5C (formerly 6C) are based on 3 independent replicates of 5 mice per treatment, this isn't clear from the text. In fact, the text and legend both repeatedly note that 5 mice were used. Are the plots based on 5 mice or 15 mice? Is there variance across the replicates? If these haven't been analyzed, then why have 3 independent replicates? This needs to be clarified. If the values are indeed only based on 5 mice/treatment, my original concern remains that these are too few to be conclusive.

Response: Thanks for your comments. We have repeated the mouse survival assay with 10 mice per treatment. In accordance with previous result, mice survival was identical for WT versus $\Delta treP$ during mono-infection; however, co-infection with *A. baumannii* augmented the virulence of WT strain, but significantly alleviated the virulence of mutant $\Delta treP$, which led to increased and prolonged mouse survival. In the revised manuscript, we have renewed the mice number as well as statistical method (please see line 557, 836, 839-840), and updated Fig. 5C based on the survival curves of 10 mice per treatment. Thanks for your understanding.

We have tried our best to improve the manuscript and hope we have addressed all the questions raised. We appreciate your earnest work, and hope that the corrections will meet with approval.

Once again, thank you very much for your professional comments and suggestions.

Looking forward to hearing from you.

Best regards.

Yours sincerely,

Yan Zhao, PhD

Department of Microbiology,
Army Medical University,
30# Gaotanyan St., Shapingba District,
Chongqing 400038, China
Tel: 86-23-68752243
E-mail: hnyanyanxp@aliyun.com

August 13, 2022

Dr. Yan Zhao
Third Military Medical University
Department of Microbiology
gaotanyan street #30
chongqing 400038
China

Re: mSystems00338-22R2 (Essential fitness repertoire of *Staphylococcus aureus* during co-infection with *Acinetobacter baumannii in vivo*)

Dear Dr. Yan Zhao:

Your manuscript has been accepted, and I am forwarding it to the ASM Journals Department for publication. For your reference, ASM Journals' address is given below. Before it can be scheduled for publication, your manuscript will be checked by the mSystems production staff to make sure that all elements meet the technical requirements for publication. They will contact you if anything needs to be revised before copyediting and production can begin. Otherwise, you will be notified when your proofs are ready to be viewed.

Publication Fees:

If you would like to submit a potential Featured Image, please email a file and a short legend to msystems@asmusa.org. Please note that we can only consider images that (i) the authors created or own and (ii) have not been previously published. By submitting, you agree that the image can be used under the same terms as the published article. File requirements: square dimensions (4" x 4"), 300 dpi resolution, RGB colorspace, TIF file format.

We recognize that the video files can become quite large, and so to avoid quality loss ASM suggests sending the video file via <https://www.wetransfer.com/>. When you have a final version of the video and the still ready to share, please send it to mSystems staff at msystems@asmusa.org.

Sincerely,

Gilles van Wezel
Editor, mSystems

Journals Department
Supplemental Table 3: Accept
Supplemental Table 2: Accept
Supplemental Table 4: Accept
Supplemental Table 1: Accept